# Insight into the Causal Relationship between Gut Microbiota and Back Pain: A Two Sample Bidirectional Mendelian Randomization Study

*Jingni Hui, Yujing Chen, Chun'e Li, Yifan Gou, Ye Liu, Ruixue Zhou, Meijuan Kang, Chen Liu, Bingyi Wang, Panxing Shi, Shiqiang Cheng, Xuena Yang, Chuyu Pan, Yumeng Jia, Bolun Cheng, Huan Liu, Yan Wen,\* and Feng Zhang\**

Observational studies have shown that alterations in gut microbiota composition are associated with low back pain. However, it remains unclear whether the association is causal. To reveal the causal association between gut microbiota and low back pain, a two-sample bidirectional Mendelian randomization (MR) analysis is performed. The inverse variance weighted regression (IVW) is performed as the principal MR analysis. MR-Egger and Weighted Median is further conducted as complementary analysis to validate the robustness of the results. Finally, a reverse MR analysis is performed to evaluate the possibility of reverse causation. The inverse variance weighted (IVW) method suggests that *Peptostreptococcaceae* (odds ratio [OR] 1.056, 95% confidence interval [CI] [1.015–1.098], $P_{IVW}$ = 0.010), and *Lactobacillaceae* (OR 1.070, 95% CI [1.026–1.115], $P_{IVW}$ = 0.003) are positively associated with back pain. The *Ruminococcaceae* (OR 0.923, 95% CI [0.849–0.997], $P_{IVW}$ = 0.033), *Butyricicoccus* (OR 0.920, 95% CI [0.868 - 0.972], $P_{IVW}$ = 0.002), and *Lachnospiraceae* (OR 0.948, 95% CI [0.903–0.994], $P_{IVW}$ = 0.022) are negatively associated with back pain. In this study, underlying causal relationships are identified among gut microbiota and low back pain. Notably, further research is needed on the biological mechanisms by which gut microbiota influences low back pain.

J. Hui, Y. Chen, C. Li, Y. Gou, Y. Liu, R. Zhou, M. Kang, C. Liu, B. Wang, P. Shi, S. Cheng, X. Yang, C. Pan, Y. Jia, B. Cheng, H. Liu, Y. Wen, F. Zhang
Key Laboratory of Trace Elements and Endemic Diseases of National Health and Family Planning Commission
School of Public Health
Health Science Center
Xi'an Jiaotong University
Xi'an 71006, P. R. China
E-mail: wenyan@mail.xjtu.edu.cn; fzhxjtu@mail.xjtu.edu.cn

## 1. Introduction

The Global Burden of Disease Study in 2019 showed that back pain (BP) is one of the leading causes of years of disability loss.[1] It causes severe discomfort and disability to individuals and imposes a considerable economic and health-care burden on society. The prevalence of back pain was 33.9% among adults,[2] and a total lifetime prevalence was 47%.[3] Due to the high prevalence and heavy burden of back pain globally, it is critical to discover modifiable risk factors. The burden of back pain is increasing with aging, and new treatment strategies are urgently needed to prevent and alleviate back pain.[4]

Back pain is the result of a combination of biological, psychological, and social factors. Previous population studies have found that intervertebral disc disease (IDD) is a major cause of back pain. Back pain has a significant genetic susceptibility, with an estimated heritability of between 30% and 68%.[5–7] Recently, the genome-wide association studies (GWAS) have identified accumulated genetic loci.[8–11] A large-scale GWAS study of over 500 000 individuals identified a common underlying genetic component between disc pathology and low back pain, identifying three BP-associated loci.[8] Another large-scale genome-wide association study meta-analysis identified 41 variants at 33 motifs associated with back pain.[11] These findings provide new insights into the etiology of back pain from a genetic perspective.

The human gastrointestinal tract is home to the most complex human microbial ecosystem. The gut microbiota is involved in regulating and maintaining the host's immune system.[12] Environmental factors such as diet and medications can lead to dysbiosis of the gut microbiota, which can cause a reduction in the integrity and function of the intestinal barrier, thus promoting chronic inflammation and inducing pain.[13] Increasing evidence showed that ecological dysbiosis of the gut microbiota is associated with various osteoarticular diseases. Observational studies

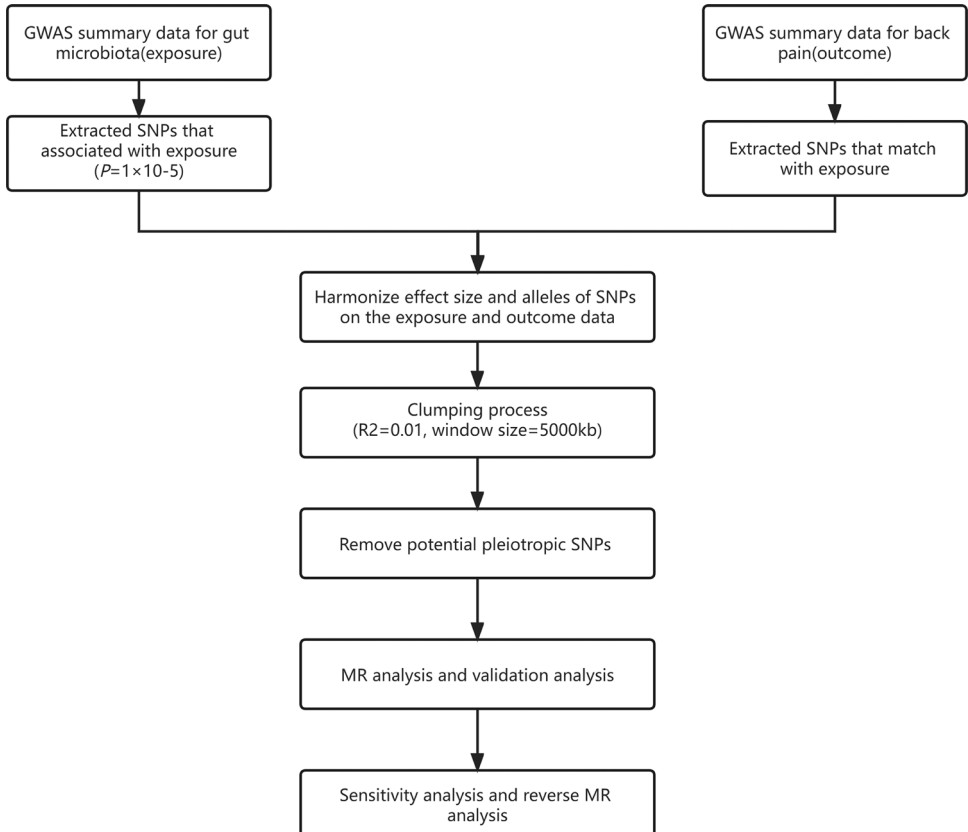

**Figure 1.** The study design of the bidirectional MR study. MR, Mendelian randomization; SNP, single nucleotide polymorphisms.

have shown that the composition of the gut microbiota in patients with osteoporosis, osteoarthritis, and rheumatoid arthritis has altered. Osteoarthrosis (OA) is associated with higher levels of *Streptococcus*[14,15] and *Bacteroide*.[16] Several studies have demonstrated the association between microbiota dysbiosis and rheumatoid arthritis (RA).[17] Numerous studies have shown that the gut microbiota plays a very important role in the prevention of bone loss.[18] *Dekker Nitert* et al. found that low back pain in overweight individuals was associated with alterations in the composition of the gut microbiota, particularly the higher relative abundance of *Adlercreutzia*, *Roseburia*, and *Christensenellaceae* in overweight individuals with low back pain.[19] Nevertheless, the causal relationships between the gut microbiota and back pain remain unclear.

Mendelian randomization is a genetic epidemiological method to investigate whether there is a causal effect between exposure and outcome variables, dealing with confounding factors. Genetic variants significantly associated with exposure are selected as instrumental variables to infer the causal effect of exposure on outcome. Genetic variants should be satisfied three basic conditions[20]: 1) the genetic variation is related to the exposure; 2) the genetic variation affects with the outcome merely through the exposure; 3) the genetic variation is independent of any confounders. Mendelian randomization (MR) was used to evaluate the causal relationship between gut microbiota and back pain.

## 2. Experimental Section

### 2.1. Study Design

**Figure 1** shows the flowchart of this MR study. A two-sample bidirectional MR analysis was performed from two independent populations. In the forward MR study, the gut microbiota was set as the exposure, whereas back pain was determined as the outcome, respectively. In reverse MR, back pain was set as exposure, and the gut microbiota was set as the outcome. The study used publicly available summary statistics for analysis, and no ethical approval was required.

### 2.2. Genome-Wide Association Study Summary Data for Back Pain

GWAS summary statistics for back pain were derived from a previous study[11] containing 1 028 947 cases of dorsalgia (119 100 cases, 909 847 controls) and 981 812 cases of back pain due to IDD (58 854 cases, 922 958 controls). This study focused on two of the most common physician-assigned back pain diagnosed as defined under the International Statistical Classification of Diseases (ICD-10) that are IDD (code M51) and dorsalgia (code M54); representing largely known (IDD) and unknown (dorsalgia) etiologies of back pain. Meta-analysis of GWASs was from Iceland (deCODE Genetics), Denmark

(Danish Blood Donor Study; DBDS and Copenhagen Hospital Biobank; CHB), the United Kingdom (UK Biobank; UKB), and Finland (FinnGen). All subjects were of European descent. For deCODE Genetics, they had sequenced whole genomes of 49 962 Icelanders using GAIIx, HiSeq, HiSeqX, and NovaSeq Illumina technology to a mean depth of at least 17.8×. Chip-typing of Danish samples was performed using the Illumina Infinium Global Screening Array. In total, over 332 000 samples from the CHB and DBDS, together with ≈238000 genotyped samples from Northwestern Europe were long-range phased using Eagle2. Samples and variants with <98% yield were excluded. Samples of UKB participants were genotyped with a custom-made Affymetrix chip, UK BiLEVE Axiom, in the first 50 000 individuals, and the Affymetrix UK Biobank Axiom array in the remaining participants. A total of ≈38.0 million variants were analyzed in the UKB dataset. For FinnGen, Genotyping was performed on the ThermoFisher Axiom array in San Diego. Genotype calls were made with the AxiomGT1 algorithm. Imputation was performed using the Finnish population-specific and high-coverage WGS backbone and the population-specific SISu v3 imputation reference panel with Beagle 4.1. A total of 14.5 million variants were analyzed in the Finnish dataset. Cross-trait LD-score regression and summary statistics from traits in the deCODE and UKB datasets or available meta-analyses were used in the previous studies. Detailed information about methods, genotyping, quality control, imputation, process, and approaches can be found in the previous studies.[11]

### 2.3. Genome-Wide Association Study Summary Data for Gut Microbiota

GWAS summary statistics related to gut microbiota obtained from a multi ancestry large-scale GWAS meta-analysis including 18340 individuals from 24 cohorts by 16S rRNA sequencing.[21] All participants were from the United States, Canada, Israel, South Korea, Germany, Denmark, the Netherlands, Belgium, Sweden, Finland and the United Kingdom. Twenty-three of 24 cohorts used the Michigan Imputation Server for imputation, using the HRC 1.0 or 1.1 reference panel. They selected the best matching 1000 SNPs for each top SNP using SNPSNAP, matched by allele frequency, gene density, number of LD pairs, and distance from the closest gene. This study used the genome-wide threshold for significance of $P < 5 \times 10^{-8}$. This study yielded a total of 211 taxa including 131 genera, 35 families, 20 orders, 16 classes, and 9 phyla. Then association analysis was performed by Spearman correlation with covariates such as age, sex, technical covariates, and genetic PCs. Detailed information about Experimental Section, genotyping, quality control, imputation, process, and approaches could be found in the previous studies.[22]

### 2.4. Validation Analysis

A validation analysis was conducted to verify the results from first-step MR analysis results. In this step, it used another public GWAS summary data of back pain. Summary statistics of back pain could be downloaded from www.ebi.ac.uk/gwas (GCST90044406 and GCST90043767). The validation outcome

samples for back pain were obtained from the UK Biobank study (UKB data field 6159_4 and 3157), which was a large prospective cohort study with over 500 000 participants aged 40–69 years from 22 centers across the United Kingdom. Detailed information could be found in the original study.[23]

### 2.5. Statistical Analysis

Bidirectional MR analysis was conducted to estimate the causal relationship between gut microbiota and back pain. The gut microbiota was selected as instrumental variables (IVs) with a genome-wide threshold of $1 \times 10^{-5}$. We conducted several quality-control measures to select qualified IVs. In this study, SNPs were first excluded with inconsistent alleles between exposure and outcome (i.e., A/G versus A/C). Second, the linkage disequilibrium may lead to biased results. Based on the European 1000 Genomes Project reference panel, LD between included SNPs was assessed using clumping process among genome-wide SNPs (clump_r2 = 0.01, clump_kb = 5000, $P = 1 \times 10^{-5}$). The inverse variance weighted regression (IVW)[24] was performed as the principal MR analysis. IVW is an effective analysis to infer the causality between exposure and outcome when all genetic variants are valid IVs. In the absence of horizontal pleiotropy, the IVW method will obtain an unbiased estimate.[25] MR-Egger[26] and Weighted Median[27] were further considered as complementary methods to enhance the reliability and robustness of the estimate. The Q test in IVW and MR-Egger regression were conducted to evaluate the potential heterogeneity. $P > 0.05$ indicates that there was no heterogeneity in the included instrument variables. MR-Egger intercept was used to evaluate whether the included SNPs had the potential horizontal pleiotropy. If $P > 0.05$, it was considered that there was no horizontal pleiotropy. The weighted median provides consistent estimates when at least 50% of the information were valid instrument variables. Finally, MR-PRESSO[28] was used to estimate the pleiotropy and corrects for it by removing outliers from the IVW model.

Reverse MR analysis was performed to explore whether back pain had any causal effects on the identified significant bacterial genus. Briefly, back pain (BP_dorsalgia and BP_IDD) served as exposure and the identified causal microbiome was outcome. The reverse MR selected SNPs significantly associated with back pain with a genome-wide threshold of $5 \times 10^{-8}$.

All statistical analyses were implemented by R (version 4.2.1). The IVW, weighted median, and MR-Egger regression methods were conducted using the "TwoSampleMR" package (version 4.2.1). The MR-PRESSO test was performed using the "MR-PRESSO" package.

## 3. Results

### 3.1. Causal Effect of Gut Microbiota on Back Pain (BP_ Dorsalgia)

**Table 1** demonstrated the causal relationship between gut microbiota and dorsalgia. We observed that 12 taxa (1 class, 3 families, 7 genuses, and 1 order) were the risk factors for BP_dorsalgia. We found that the genetically predicated Class *Deltaproteobacteria* was negatively associated with BP_dorsalgia (odds ratio [OR]

**Table 1.** Mendelian randomization results of gut microbiota on back pain.

| Exposure | Outcome | Method | Nsnp | p | OR[95% CI] | Heterogeity | Pleiotropy | MR-PRESSO |
|---|---|---|---|---|---|---|---|---|
| class.Deltaproteobacteria | BP_Dorsalgia | MR Egger | 12 | 0.689 | 0.950(0.744–1.213) | 0.409 | 0.979 | 0.239 |
| | BP_Dorsalgia | Weighted median | 12 | 0.045 | 0.926(0.858–0.998) | | | |
| | BP_Dorsalgia | IVW | 12 | 0.035 | 0.947(0.896–0.998) | 0.498 | | |
| family.Peptostreptococcaceae | BP_Dorsalgia | MR Egger | 11 | 0.419 | 1.041(0.949–1.142) | 0.440 | 0.733 | 0.229 |
| | BP_Dorsalgia | Weighted median | 11 | 0.022 | 1.072(1.010–1.137) | | | |
| | BP_Dorsalgia | IVW | 11 | 0.010 | 1.056(1.015–1.098) | 0.523 | | |
| family.Ruminococcaceae | BP_Dorsalgia | MR Egger | 8 | 0.618 | 1.033(0.916–1.163) | 0.334 | 0.076 | 0.041 |
| | BP_Dorsalgia | Weighted median | 8 | 0.236 | 0.949(0.871–1.035) | | | |
| | BP_Dorsalgia | IVW | 8 | 0.033 | 0.923(0.849–0.997) | 0.098 | | |
| family.unknownfamily | BP_Dorsalgia | MR Egger | 12 | 0.402 | 1.091(0.897–1.327) | 0.002 | 0.912 | 0.006 |
| | BP_Dorsalgia | Weighted median | 12 | 0.148 | 1.046(0.984–1.111) | | | |
| | BP_Dorsalgia | IVW | 12 | 0.018 | 1.080(1.016–1.143) | 0.004 | | |
| genus.Eubacterium fissicatena group | BP_Dorsalgia | MR Egger | 9 | 0.924 | 1.010(0.836–1.220) | 0.150 | 0.790 | 0.273 |
| | BP_Dorsalgia | Weighted median | 9 | 0.176 | 1.029(0.987–1.071) | | | |
| | BP_Dorsalgia | IVW | 9 | 0.036 | 1.037(1.003–1.070) | 0.209 | | |
| genus.Allisonella | BP_Dorsalgia | MR Egger | 8 | 0.545 | 0.941(0.781–1.134) | 0.810 | 0.325 | 0.752 |
| | BP_Dorsalgia | Weighted median | 8 | 0.148 | 1.029(0.990–1.070) | | | |
| | BP_Dorsalgia | IVW | 8 | <0.001 | 1.041(1.020–1.062) | 0.764 | | |
| genus.Butyricicoccus | BP_Dorsalgia | MR Egger | 6 | 0.157 | 0.707(0.478–1.045) | 0.948 | 0.251 | 0.779 |
| | BP_Dorsalgia | Weighted median | 6 | 0.135 | 0.931(0.847–1.023) | | | |
| | BP_Dorsalgia | IVW | 6 | 0.002 | 0.920(0.868–0.972) | 0.772 | | |
| genus.Collinsella | BP_Dorsalgia | MR Egger | 9 | 0.236 | 0.871(0.708–1.073) | 0.602 | 0.608 | 0.851 |
| | BP_Dorsalgia | Weighted median | 9 | 0.021 | 0.912(0.844–0.986) | | | |
| | BP_Dorsalgia | IVW | 9 | 0.001 | 0.920(0.872–0.969) | 0.674 | | |
| genus.Escherichia.Shigella | BP_Dorsalgia | MR Egger | 10 | 0.387 | 0.940(0.823–1.073) | 0.589 | 0.973 | 0.769 |
| | BP_Dorsalgia | Weighted median | 10 | 0.058 | 0.940(0.881–1.002) | | | |
| | BP_Dorsalgia | IVW | 10 | 0.003 | 0.942(0.902–0.981) | 0.687 | | |
| genus.RuminococcaceaeUCG004 | BP_Dorsalgia | MR Egger | 9 | 0.619 | 0.906(0.625–1.314) | 0.032 | 0.870 | 0.090 |
| | BP_Dorsalgia | Weighted median | 9 | 0.523 | 0.979(0.916–1.045) | | | |
| | BP_Dorsalgia | IVW | 9 | 0.037 | 0.935(0.872–0.998) | 0.051 | | |
| genus.RuminococcaceaeUCG013 | BP_Dorsalgia | MR Egger | 12 | 0.521 | 0.956(0.838–1.091) | 0.428 | 0.942 | 0.496 |
| | BP_Dorsalgia | Weighted median | 12 | 0.163 | 0.953(0.891–1.020) | | | |
| | BP_Dorsalgia | IVW | 12 | 0.034 | 0.952(0.906–0.997) | 0.517 | | |
| order.MollicutesRF9 | BP_Dorsalgia | MR Egger | 12 | 0.402 | 1.091(0.897–1.327) | 0.002 | 0.912 | <0.001 |
| | BP_Dorsalgia | Weighted median | 12 | 0.157 | 1.046(0.983–1.112) | | | |
| | BP_Dorsalgia | IVW | 12 | 0.018 | 1.080(1.016–1.143) | 0.004 | | |
| class.Deltaproteobacteria | BP_IDD | MR Egger | 12 | 0.635 | 0.922(0.664–1.278) | 0.498 | 0.922 | 0.183 |
| | BP_IDD | Weighted median | 12 | 0.069 | 0.913(0.827–1.007) | | | |
| | BP_IDD | IVW | 12 | 0.003 | 0.907(0.842–0.971) | 0.587 | | |
| family.Desulfovibrionaceae | BP_IDD | MR Egger | 9 | 0.895 | 0.977(0.698–1.367) | 0.506 | 0.704 | 0.222 |
| | BP_IDD | Weighted median | 9 | 0.088 | 0.912(0.820–1.014) | | | |
| | BP_IDD | IVW | 9 | 0.013 | 0.914(0.843–0.985) | 0.597 | | |
| family.Lactobacillaceae | BP_IDD | MR Egger | 9 | 0.271 | 1.081(0.951–1.228) | 0.503 | 0.874 | 0.327 |
| | BP_IDD | Weighted median | 9 | 0.031 | 1.076(1.007–1.151) | | | |
| | BP_IDD | IVW | 9 | 0.003 | 1.070(1.026–1.115) | 0.609 | | |
| family.Veillonellaceae | BP_IDD | MR Egger | 17 | 0.803 | 1.017(0.893–1.159) | 0.375 | 0.185 | 0.164 |
| | BP_IDD | Weighted median | 17 | 0.178 | 0.949(0.880–1.024) | | | |
| | BP_IDD | IVW | 17 | 0.021 | 0.935(0.879–0.992) | 0.314 | | |

*(Continued)*

**Table 1.** (Continued).

| Exposure | Outcome | Method | Nsnp | p | OR[95% CI] | Heterogeity | Pleiotropy | MR-PRESSO |
|---|---|---|---|---|---|---|---|---|
| genus.Allisonella | BP_IDD | MR Egger | 8 | 0.960 | 1.007(0.762–1.332) | 0.299 | 0.802 | 0.506 |
| | BP_IDD | Weighted median | 8 | 0.191 | 1.034(0.983–1.088) | | | |
| | BP_IDD | IVW | 8 | 0.024 | 1.045(1.007–1.084) | 0.396 | | |
| genus.Hungatella | BP_IDD | MR Egger | 5 | 0.632 | 1.103(0.768–1.586) | 0.355 | 0.861 | 0.591 |
| | BP_IDD | Weighted median | 5 | 0.018 | 1.094(1.016–1.178) | | | |
| | BP_IDD | IVW | 5 | 0.014 | 1.066(1.015–1.116) | 0.511 | | |
| genus.LachnospiraceaeUCG001 | BP_IDD | MR Egger | 13 | 0.481 | 0.919(0.733–1.153) | 0.602 | 0.786 | 0.438 |
| | BP_IDD | Weighted median | 13 | 0.157 | 0.950(0.885–1.020) | | | |
| | BP_IDD | IVW | 13 | 0.022 | 0.948(0.903–0.994) | 0.678 | | |
| genus.Lactobacillus | BP_IDD | MR Egger | 8 | 0.237 | 1.103(0.953–1.276) | 0.372 | 0.632 | 0.485 |
| | BP_IDD | Weighted median | 8 | 0.053 | 1.073(0.999–1.152) | | | |
| | BP_IDD | IVW | 8 | 0.015 | 1.065(1.014–1.115) | 0.455 | | |
| genus.Marvinbryantia | BP_IDD | MR Egger | 10 | 0.330 | 1.151(0.882–1.501) | 0.383 | 0.650 | 0.325 |
| | BP_IDD | Weighted median | 10 | 0.185 | 1.064(0.971–1.166) | | | |
| | BP_IDD | IVW | 10 | 0.018 | 1.082(1.017–1.148) | 0.459 | | |
| genus.RuminococcaceaeUCG003 | BP_IDD | MR Egger | 11 | 0.776 | 0.970(0.788–1.192) | <0.001 | 0.714 | 0.001 |
| | BP_IDD | Weighted median | 11 | 0.097 | 0.927(0.848–1.014) | | | |
| | BP_IDD | IVW | 11 | 0.003 | 0.927(0.876–0.978) | <0.001 | | |
| order.MollicutesRF9 | BP_IDD | MR Egger | 12 | 0.815 | 1.025(0.837–1.256) | 0.121 | 0.676 | 0.406 |
| | BP_IDD | Weighted median | 12 | 0.239 | 1.049(0.969–1.137) | | | |
| | BP_IDD | IVW | 12 | 0.049 | 1.069(1.003–1.135) | 0.156 | | |
| order.Rhodospirillales | BP_IDD | MR Egger | 14 | 0.133 | 1.197(0.962–1.489) | 0.503 | 0.049 | 0.044 |
| | BP_IDD | Weighted median | 14 | 0.136 | 0.949(0.886–1.017) | | | |
| | BP_IDD | IVW | 14 | 0.031 | 0.943(0.889–0.996) | 0.243 | | |

Nsnp: Number of SNPs involved in the analysis; *P*: p value of effect estimate; OR: odds ratio; CI: confidence interval; IVW: Inverse variance weighted.

0.947, 95% confidence interval [CI] [0.896 - 0.998], $P_{IVW}$ = 0.035). In the family level, the *Peptostreptococcaceae* was positively associated with BP_dorsalgia (OR 1.056, 95% CI [1.015–1.098], $P_{IVW}$ = 0.010), and the *Ruminococcaceae* was negatively associated with BP_dorsalgia (OR 0.923, 95% CI [0.849–0.997], $P_{IVW}$ = 0.033). In genus level, the *Eubacterium fissicatena group* (OR 1.037, 95% CI [1.003–1.070], $P_{IVW}$ = 0.036) and *Allisonella* (OR 1.041, 95% CI [1.020–1.062], $P_{IVW}$ <0.001) were positively associated with BP_dorsalgia. The *Butyricicoccus* (OR 0.920, 95% CI [0.868–0.972], $P_{IVW}$ = 0.002), *Collinsella* (OR 0.920, 95% CI [0.872–0.969], $P_{IVW}$ = 0.001), *Escherichia* (OR 0.942, 95% CI [0.902–0.981], $P_{IVW}$ = 0.003), were negatively associated with BP_dorsalgia. In order level, the *MollicutesRF9* (OR 1.080, 95% CI [1.016–1.143], $P_{IVW}$ = 0.018) was positively associated with BP_dorsalgia.

### 3.2. Causal Effect of Gut Microbiota on Back Pain (BP_IDD (Intervertebral Disc Disease))

Table 1 demonstrated the causal relationship between gut microbiota and back pain due to IDD. We observed that 12 taxa (1 class, 3 families, 6 genuses, and 2 orders) were the risk factors for BP_dorsalgia. We found that the genetically predicated Class *Deltaproteobacteria* was negatively associated with BP_IDD (OR 0.907, 95% CI [0.842–0.971], $P_{IVW}$ = 0.003). In the family level, the *Lactobacillaceae* was positively associated with BP_IDD (OR 1.070, 95% CI [1.026–1.115], $P_{IVW}$ = 0.003), and the *Desulfovib-*

*rionaceae* (OR 0.914, 95% CI [0.843–0.985], $P_{IVW}$ = 0.013) and *Veillonellaceae* (OR 0.935, 95% CI [0.879–0.992], $P_{IVW}$ = 0.021) was negatively associated with BP_IDD. In genus level, the *Allisonella* (OR 1.045, 95% CI [1.007–1.084], $P_{IVW}$ = 0.024), *Hungatella* (OR 1.066, 95% CI [1.015–1.116], $P_{IVW}$ = 0.014) and *Marvinbryantia* (OR 1.082, 95% CI [1.017–1.148], $P_{IVW}$ = 0.018) were positively associated with BP_IDD. The *LachnospiraceaeUCG001* (OR 0.948, 95% CI [0.903–0.994], $P_{IVW}$ = 0.022) and *RuminococcaceaeUCG003* (OR 0.927, 95% CI [0.876–0.978], $P_{IVW}$ = 0.003) were negatively associated with BP_IDD. In order level, the *Rhodospirillales* (OR 0.943, 95% CI [0.889–0.996], $P_{IVW}$ = 0.031) were negatively associated with BP_IDD.

The causal effects of the *Veillonellaceae* (OR 0.961, 95% CI [0.922–1.000], $P_{IVW}$ = 0.045) on back pain were successfully validated, as shown in Table S1 (Supporting Information). The effect direction was consistent with that in the previous sample, which strengthened the confidence of the true causal associations.

### 3.3. Reverse Mendelian Randomization of Back Pain on Gut Microbiota

In the reverse MR, we did not detect significant causal effects of back pain on gut microbiota detected by the forward MR analysis. However, we found evidence that back pain was significantly associated with the other four gut microbiota (**Table 2**). Dorsalgia was positively with the genetically predicated genus *Methanobre-*

**Table 2.** The results of reverse Mendelian randomization.

| Exposure | Outcome | Method | Nsnp | p | OR(95% CI) | Heterogeity | Pleiotropy | MR-PRESSO |
|---|---|---|---|---|---|---|---|---|
| BP_Dorsalgia | genus.Methanobrevibacter | MR Egger | 18 | 0.407 | 0.368(0.037–3.687) | 0.841 | 0.257 | 0.858 |
| | | Weighted median | 18 | 0.231 | 1.360(0.822–2.251) | | | |
| | | IVW | 18 | 0.018 | 1.440(1.137–1.742) | 0.809 | | |
| BP_Dorsalgia | order.Rhodospirillales | MR Egger | 18 | 0.534 | 1.664(0.346–7.992) | 0.323 | 0.740 | 0.229 |
| | | Weighted median | 18 | 0.138 | 1.284(0.923–1.787) | | | |
| | | IVW | 18 | 0.048 | 1.275(1.034–1.515) | 0.379 | | |
| BP_Dorsalgia | phylum.Euryarchaeota | MR Egger | 18 | 0.437 | 0.404(0.043–3.752) | 0.926 | 0.282 | 0.925 |
| | | Weighted median | 18 | 0.040 | 1.639(1.024–2.625) | | | |
| | | IVW | 18 | 0.011 | 1.412(1.145–1.679) | 0.907 | | |
| BP_IDD | genus.LachnospiraceaeFCS020group | MR Egger | 37 | 0.198 | 0.829(0.626–1.097) | 0.502 | 0.587 | 0.243 |
| | | Weighted median | 37 | 0.082 | 0.900(0.800–1.013) | | | |
| | | IVW | 37 | 0.006 | 0.893(0.812–0.974) | 0.535 | | |

Nsnp, number of SNPs involved in the analysis; p, p-value of effect estimate; OR, odds ratio; CI, confidence interval; IVW, Inverse variance weighted.

vibacter (OR 1.440, 95% CI [1.137–1.742], $P_{IVW}$ = 0.018), order *Rhodospirillales* (OR 1.275, 95% CI [1.034–1.515], $P_{IVW}$ = 0.048) and phylum *Euryarchaeota* (OR 1.412, 95% CI [1.145–1.679], $P_{IVW}$ = 0.011). Back pain due to intervertebral disc disorder was negatively with genus *LachnospiraceaeFCS020* (OR 0.893, 95% CI [0.812–0.974], $P_{IVW}$ = 0.006).

### 3.4. Sensitivity Analysis

We performed a sensitivity analysis to verify our putative causal relationships obtained with bidirectional MR. First, the global MR pleiotropy residual sum and outlier (MR-PRESSO global test) suggested that there was significant horizontal pleiotropy between the instrumental variables of *Ruminococcaceae* (P = 0.041), *RuminococcaceaeUCG003* (P = 0.001), *Rhodospirillales* (P = 0.044) and outcome (Table 1). Second, MR-Egger results showed there was evidence of horizontal pleiotropy between *Rhodospirillales* (P = 0.049) and back pain due to IDD. Third, The Q test in IVW and MR-Egger method showed there was evidence of heterogeneity between the instrumental variables of *RuminococcaceaeUCG004* ($P_{MR-Egger}$ = 0.032), *MollicutesRF9* ($P_{MR-Egger}$ = 0.002, $P_{IVW}$ = 0.004), *RuminococcaceaeUCG003*($P_{MR-Egger}$ <0.001, $P_{IVW}$ <0.001) and outcome. Fourth, the directions of the estimates from another two MR methods were the same as those of the IVW method. However, using the same threshold (P <1 × 10⁻⁵), the numbers of associations supported by the weighted median and MR-Egger method were only 5 and 0, respectively (Table 1). The difference may be due to the fact that the power of these two methods is smaller than that of the IVW.[29] Overall, the sensitivity analysis confirmed the reliability of our putative causal effects in both the forward and reverse MR results.

### 4. Discussion

Although growing evidence showed the association between gut microbiota and skeletal system disorders, there were few studies on the relationship between gut microbiota and low back pain. In this study, a two-sample bidirectional MR analysis was performed to investigate the causal relationship between gut microbiota and back pain. We observed several genetically determined gut microbiota are potentially associated with the risk of back pain (11 taxa associated with BP_dorsalgia and 12 taxa associated with BP_IDD).

Several studies have observed a significant imbalance between pro-inflammatory cytokines and anti-inflammatory factors in patients with back pain. Patients with back pain had higher levels of proinflammatory cytokines, and serum mRNA levels of interleukin-6 (IL-6), IL-8,[30] and tumor necrosis fator TNF-α.[31] Anti-inflammatory cytokines such as IL-4[31] and IL-10 have lower levels in patients with back pain. In this study, we observed *Peptostreptococcaceae* were positively associated with the risk of back pain. An animal study[32] showed that *Peptostreptococcaceae* was increased in the dextran sulfate sodiun-induced colitis mouse model and exhibited increases in pro-inflammatory cytokines (interleukin IL-1β, IL-6, and tumor necrosis factor TNF-α) and decreases in an anti-inflammatory cytokine (IL-13) in the serum. IL-1[33] plays a prominent role in both cartilage degradation and stimulation of nociceptive pathways. IL-1β[34] has been shown to activate nociceptors. IL-6[35,36] is involved in cartilage degradation, and has also been associated with hyperalgesia and hypersensitivity in joint tissues. TNF-α is also involved in cartilage destruction[37] and activation of nociceptors.[38] *Peptostreptoccaceae* may play a positive role to cause back pain by promoting the elevation of proinflammatory factors such as IL-1 β, IL-6, and TNF-α in vivo. A study[39] found that *Lactobacillus acidophilus* improved monosodium iodoacetate (MIA)-induced OA development through downregulation of pain severity and cartilage damage. *Lactobacillus acidophilus* decreased the expression of proinflammatory cytokines but promoted anti-inflammatory cytokines production in joint of OA rat model. *Lactobacillus acidophilus* regulated a balance between anabolic and catabolic factors in chondrocytes from OA patients. However, in this study, *Lactobacillus* was positively associated with the risk of low back pain. Further studies should be conducted to investigate the mechanisms among *Lactobacillus* and the risk of low back pain.

In addition, bidirectional MR analysis identified *Ruminococcaceae* and *Lachnospiraceae* were negatively associated with back

pain. *Lachnospiraceae* and *Ruminococcaceae* belongs to *Firmicutes*. A metagenomic[40] study demonstrated a significant decrease in the abundance of *Firmicutes*, *Actinobacteria*, and *Bacteroidetes* in diseased discs. An animal study[41] found higher relative abundance of *Lachnospiraceae* and *Ruminococcaceae* and lower relative abundance of *Lactobacillus* in lumbar disc herniation (LDH) mice treated with *Lactobacillus s16*. One study[42] suggests that *Lycium barbarum* polysaccharide alleviated RA by reshaping the composition of intestinal microbiota. Specifically, *Lycium barbarum* polysaccharide intervention reduced the relative abundance of *Lachnospiraceae* and *Ruminococcaceae*. Gut bacterial genera *Lachnospiraceae* and *Ruminococcus* are associated with both prevalent musculoskeletal pain (MSKP) and a higher number of MSKP locations in older community-dwelling men.[43]

*Lachnospiraceae* and *Ruminococcaceae* are key components of the healthy human microbiota by maintaining the homeostasis of the intestinal microenvironment. Accumulating evidence showed that *Lachnospiraceae* and *Ruminococcaceae* are associated with inflammatory diseases. An increased abundance of *Ruminococcus* has been found in patients with spondyloarthritis.[44] Breban et al. observed the abundance of *Ruminococcus* increased in spondyloarthritis compared to rheumatoid arthritis and healthy controls.[45] In a cross-sectional discovery cohort, the systemic lupus erythematosus (SLE) patients had higher abundance of *Ruminococcus gnavus* than other patients.[46] An MR study showed that *Lachnospira* was negatively associated with the risk of SLE.[25] Zhang et al. found that the abundance of *Lachnospira* and *Ruminococcus* was significantly lower in patients with ankylosing spondylitis compared to healthy controls.[47] Su et al.[48] found genetically predicted *Ruminococcaceae* were negatively correlated with the risk of back pain based on summary data from FinnGen. Accumulating evidence indicates that gut *Lachnospiraceae* and *Ruminococcaceae* were the primary producer of short chain fatty acids (SCFAs). Multiple evidence has proved that SCFAs exert anti-inflammatory role and promote the integrity of epithelial barrier function.[49,50] SCFAs, especially butyrate, also promote Th1 cell production of IL-10 so as to maintain intestinal homeostasis.[51] Furthermore, macrophages differentiated in the presence of butyrate also show enhanced antimicrobial functions.[52] Butyrate was effective in reducing the lipopolysaccharide (LPS)-induced expression of IL6 and chemokine (C-X-C motif) ligand 2 (CXCL2) in the macrophages.[53] Butyrate exerts anti-inflammatory role by inhibiting the nuclear factor kappa-light-chain-enhancer of activated B cells (NF-kappa B) signaling pathway.[54] The mechanisms by which intestinal microbiota play beneficial or detrimental roles in immune-mediated back pain remain to be further investigated. A deeper understanding of the mechanisms would benefit our intervention in the composition of the gut microbiota to prevent and relieve low back pain.

There are several advantages in our study. To the best of our knowledge, this study used a two-sample bidirectional MR analysis to explore the causal relationship between gut microbiota and low back pain based on a larger-scale summary data on back pain. Furthermore, MR results may be different from the observational studies. The MR approach reduced the interference of confounding factors and reverse causality.

However, some limitations are unavoidable in this study. First, because the participants in this study were of European ancestry, the effect of race should be considered when extrapolating relevant findings to other ethnic groups. Second, the summary data in this study were from the publicly available GWAS database, therefore, we were unable to assess the impact of population stratification on results. Further subgroup analysis was impossible because of the lack of demographics. Third, the application of a strict multiple testing correction may be conservative considering the biological plausibility and multi-stage statistical process, which may overlook potential strains causally associated with low back pain. Therefore, multiple testing was not considered in this study. Finally, horizontal pleiotropy is a common issue in MR investigation and it was no exception for this one. In this study, we applied MR-Egger and MR-PRESSO to eradicate the horizontal pleiotropy.

In conclusion, we used bidirectional MR analysis to estimate the underlying causal relationships among gut microbiota and low back pain based on large-scale GWAS summary data. We found gut *Lachnospiraceae* and *Ruminococcaceae* were negatively associated with low back pain. These findings revealed strong genetic evidence for causal links between gut microbiota and low back pain. More research is needed to evaluate whether these findings replicate in other settings and to learn more about the potential underlying mechanisms.

## Supporting Information

Supporting Information is available from the Wiley Online Library or from the author.

## Acknowledgements

The authors thank C.L., Y.C. for up-front data collation. This study was supported by the National Natural Scientific Foundation of China [81922059]; the Natural Science Basic Research Plan in Shaanxi Province of China [2021JCW-08].

## Conflict of Interests

The authors declare no conflict of interest.

## Author Contributions

J.H. and Y.C. contributed equally to this work. J.H. performed visualization, validation, and wrote-original draft preparation. F.Z. performed conceptualization, methodology, and funding acquisition. Y.C. performed validation, writing-reviewing, and editing. C.L. performed methodology and formal analysis and software. Y.G., Y.L., R.Z., M.K., C.L., B.W., P.S., S.C., X.Y., C.P., B.C., H.L., Y.W., and Y.J. performed data curation and made preparations for the manuscript at first. All authors reviewed and approved the final manuscript.

## Peer Review

The peer review history for this article is available in the Supporting Information for this article.

## Data Availability Statement

The data that support the findings of this study are available from the corresponding author upon reasonable request.

www.advancedsciencenews.com

www.advgenet.com

## Keywords

causal relationship, gut microbiota, low back pain, Mendelian randomization

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

www.advancedsciencenews.com

ADVANCED
GENETICS

www.advgenet.com

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
