## [**Supplementary Information**: Record of Transparent Peer Review · Advanced Genetics]

Record of Transparent Peer Review

Insight into the causal relationship between gut microbiota and back pain: A two Sample Bidirectional Mendelian Randomization Study

Jingni Hui*, Yujing Chen*, Chun'e Li, Yifan Gou, Ye Liu, Ruixue Zhou, Meijuan Kang, Chen Liu, Bingyi Wang, Panxing Shi, Shiqiang Cheng, Xuena Yan, Chuyu Pan, Yumeng Jia, Bolun Cheng, Huan Liu, Yan Wen#, Feng Zhang#

*The two authors contributed equally to this work

#Corresponding

Review timeline:

Data submitted: 25-Jun-2023

1st Editorial Decision: 03-Aug-2023

Revision Received: 14-Sep-2023

2nd Editorial Decision: 16-Sep-2023

Revision Received: 19-Sep-2023

Accepted: 20-Sep-2023

Editor: Yuming Hu

1st Peer Review

25-Jun-2023 to 03-Aug-2023

Reviewer #1: The authors applied two-sample MR analyses to identify causal bacterial taxa for low back pain, based on publicly available genome-wide association study summary statistics. MR analyses offered reliable evidence about causal relationships among gut microbiota and low back pain. The paper showed a scientific analyses on the problem and got a strong conclusion based on analyses. The author also mentioned the limitation of the methodology in this study. I think this paper is scientifically good for publishing after minor modifications. Here is some of my questions:

In this paper, the author stated that this is "the first 2-sample bidirectional MR analysis to explore the causal relationship between gut microbiota and low back pain based on summary data from publicly available genome-wide association studies." However, my major concern of this paper is that I found out a paper did similar analyses on the same data base. This paper is Su M, Tang Y, Kong W, Zhang S, Zhu T. Genetically supported causality between gut microbiota, gut metabolites and low back pain: a two-sample Mendelian randomization study. *Front Microbiol.* 2023 Apr 14;14:1157451. doi: 10.3389/fmicb.2023.1157451. PMID: 37125171; PMCID: PMC10140346. This *Front Microbiol* paper get the similar result based on same methodology. Can the author compare this two papers and point out what is the major difference?

Reviewer #2: Hui and Chen et al carried out a straightforward two-sample Mendelian randomization analysis to identify bidirectional causal association between gut microbiota and low back pain. They identified Peptostreptococcaceae, Lactobacillaceae as positively associated with backpain, and Ruminococcaceae, Butyricoccus, and Lachnospiraceae as negatively associated with backpain.

The presented materials were thorough and clear. However, the authors reported a long list of microbiota species with mediocre statistical significance without orthogonal evidence of causal effects or meaningful clinical implications.

Following the critical appraisal checklist for evaluating Mendelian randomisation studies by Professor George Davey Smith (Reading Mendelian randomisation studies: a guide, glossary, and checklist for clinicians. 2018;362:k601), some key questions were not satisfactorily addressed:

Core Mendelian randomisation assumptions:

The inference of Mendelian randomization studies relies heavily on core assumptions, and the violation of any of these could lead to mis-leading interpretations of the results.

v The link between the genetic variants and the proposed taxa has not been validated by other orthogonal approaches. Therefore, the relevance assumption was not well supported.

Interpretation :

v The MR results should be interpreted in the context of observational estimates and orthogonal evidence in support of a causal effect. In the discussion the authors mentioned that the MR results may be different from the observational studies. How are they different for each of the reported microbiota species? How to explain the differences?

Clinical implications:

v This is a single MR study showing the causal relationship between microbiota and backpain, can the results be validated using another set of data?

v The authors claimed that microbiota is a "modifiable" risk factor. What are the proposed interventions that could modify the microbiota in favor of less low back pain? For example, what actions could one take to decrease Peptostreptococcaceae to reduce backpain? In addition, MR provides estimates of the effects over a lifetime, will the proposed interventions at a specific age have the same sized effects?

v Why did the authors perform a reverse MR analysis? What's the clinical motivation for identifying causal effects of backpain on microbiota?

Minor comments

1. There are inconsistent uses of "2-sample" and "two-sample" in the manuscript.
2. The references for the GWAS summary data for gut microbiota are missing (Page 8, line 2).
3. I'm assuming this is a typo, but on Page 8 line 5, the instrumental variables should be the genetic variants associated with gut microbiota.

1st Editorial Decision 03-Aug-2023

Editorial Decision: Major revision after addressing the reviewers' comments

Recommendation of the reviewers

Reviewer #1 Recommends Minor Revision

Reviewer #2 Reject – unsuitable for publication anywhere

Authors' response to 1st Peer Review

14-Sep-2023

Review #1

In this paper, the author stated that this is "the first 2-sample bidirectional MR analysis to explore the causal relationship between gut microbiota and low back pain based on summary data from publicly available genome-wide association studies." However, my major concern of this paper is that I found out a paper did similar analyses on the same data base. This paper is Su M, Tang Y, Kong W,

Zhang S, Zhu T. Genetically supported causality between gut microbiota, gut metabolites and low back pain: a two-sample Mendelian randomization study. Front Microbiol. 2023 Apr 14;14:1157451. doi: 10.3389/fmicb.2023.1157451. PMID: 37125171; PMCID: PMC10140346.

1. Comments: This Front Microbiol paper get the similar result based on same methodology. Can the author compare these two papers and point out what is the major difference?

Response: Thanks for your comments.

We appreciate your kindly reminding. We are sorry to miss this latest article in our literature review. Per your guidance, we read this paper and revised our manuscript. We remove this improper description for the manuscript. After comparison, we did find some similarity in study design. We both conducted Mendelian randomization analysis to test the causality relationship of gut microbiota and back pain. But there are also some differences. The biggest difference is from the definition of back-pain phenotype. In our study, we used the summary data from the meta-analysis “Rare SLC13A1 variants associate with intervertebral disc disorder highlighting role of sulfate in disc pathology[1]” which included five European cohorts such as deCODE Genetics, Danish Blood Donor Study (DBDS) and Copenhagen Hospital Biobank (CHB), and the UK Biobank (UKB), combined with summary statistics from Finland (FinnGen). The phenotype of back pain in this article was defined as “dorsalgia” and “back pain due to intervertebral disc disorder (IDD)”. These two common physician-assigned back pain diagnoses was defined under the International Statistical Classification of Diseases (ICD-10)[2]. In the article you mentioned above, they incorporated a different back pain GWAS research when identifying associated SNPs. In this study, back pain was defined as dorsalgia and back pain due to IDD. And this GWAS data only use the cohort of FinnGen Biobank, which is part of the meta-analysis we used. Generally speaking, we used more specific phenotype. Admittedly, we share a lot of similarity in design. And we did find some common taxa to this previous result. For example, we both found genetically predicted *Ruminococcaceae* were negatively correlated with the risk of back pain. But we also found some differences in results. For example, our results showed that *Peptostreptococcaceae* were positively associated with the risk of back pain, while su et al did not found this causal relationship between *Peptostreptococcaceae* and back pain. Per your guidance, we added some discussion to compare our results. Please see the revised discussion.

Thanks!

Reviewer #2:

Major

1. Comments: The link between the genetic variants and the proposed taxa has not been validated by other orthogonal approaches. Therefore, the relevance assumption was not well supported.

Response: Thanks for your comments.

We totally understand your concern about this issue. Per your guidance, we have made a thorough review and we found some evidence to support our assumption. For example, we found *Ruminococcaceae* and *Lachnospiraceae* were negatively associated with back pain. An animal study found higher relative abundance of *Lachnospiraceae* and *Ruminococcaceae* in lumbar disc herniation (LDH) mice induced by *Lactobacillus* s16[3]. One study suggests that *Lycium barbarum* polysaccharide can alleviate rheumatoid arthritis (RA) by reshaping the composition of intestinal microbiota[4]. Specifically, *Lycium barbarum* polysaccharide intervention reduced the relative abundance of

Lachnospiraceae and *Ruminococcaceae*[4]. These studies could provide evidence that *Ruminococcaceae* and *Lachnospiraceae* are correlated with the risk of back pain. We have added the above into our discussion. Please see the revised discussion (page 14, line 16 – 19).

Thanks again!

2. Comments: The MR results should be interpreted in the context of observational estimates and orthogonal evidence in support of a causal effect. In the discussion the authors mentioned that the MR results may be different from the observational studies. How are they different for each of the reported microbiota species? How to explain the differences?

Response: Thanks for your comments.

We appreciate your professional suggestion. We have added some discussion in the revised manuscript as you suggested. Please see the following paragraphs. Dekker Nitert et al. found that low back pain in overweight individuals was associated with alterations in the gut microbiota composition, particularly the higher relative abundance of *Adlercreutzia*, *Roseburia*, and *Christensenellaceae*[5]. In our study, we identified gut *Lachnospiraceae* and *Ruminococcaceae* were negatively associated with low back pain.

Mendelian randomization is an analytic approach to assess the causality of an observed association between a modifiable exposure or risk factor and a clinically relevant outcome[6]. The MR results differs substantially from the observational association. Observational studies might suggest a positive or negative association between a specific microbiota species and back pain. MR provides estimates of causality, which implies a direction of effect. In addition, MR provides estimates of the causal effect size. It quantifies how much change in the back pain can be attributed to changes in the microbiota species. The biggest advantage of MR study is that it effectively avoids the interference of confounding factors and excludes reverse causal effects[7].

The reasons for this discrepancy between our MR study and observational study above could partly result from the following reasons. First, we focus on different study population, with our Mendelian randomization study targeting a general European population and the observational study targeting a specific population. Second, the sample sizes of the studies were different, with the Mendelian study design incorporating a larger sample size. In our study, we used the largest genetic study of back pain phenotypes to date; dorsalgia (119,100 cases, 909,847 controls) and intervertebral disc disorder (IDD) (58,854 cases, 922,958 controls).

Thanks again!

3. Comments: This is a single MR study showing the causal relationship between microbiota and back pain, can the results be validated using another set of data?

Response: Thanks for your comments.

We totally understand your concern. We agree with you that validation step is necessary for MR analysis. So we carefully searched previous literature and finally found another GWAS data about back pain, which is from the UK Biobank population(www.ebi.ac.uk/gwas) [8]. We used this dataset to conduct a repeated analysis for the significant bacterial taxa identified by previous analysis. According to the validation results, *Veillonellaceae* was found to be negatively correlated with the risk of back pain, which was consistent to the first-step analysis.

In this revised manuscript, we have modified the methods and results section to add a validation analysis as you suggested. Please see line 4 - 12 in page 8 and line 13 - 16 in page 11.

Thanks again!

4. Comments: The authors claimed that microbiota is a "modifiable" risk factor. What are the proposed interventions that could modify the microbiota in favor of less low back pain? For example, what actions could one take to decrease *Peptostreptococcaceae* to reduce back pain?

Response: Thanks for your comments.

Some studies have suggested that imbalances in the gut microbiota may be associated with certain chronic disorders. These associations are not necessarily caused by a certain specie but by broader changes in the gut microbiota composition which is closely related to the inflammation bioprocesses and the nervous system. Gut microbiota may exert important anti-inflammatory effects through the production of SCFAs (acetate, butyrate and propionate) and other microbial metabolites that restore normal gut permeability[9]. Based on these mechanisms, targeting gut microbiota through dietary intervention, vitamins, trace elements, probiotics supplementation or faecal microbiota transplantation, might be a novel and effective strategy for back pain management[10].

The relationship between gut microbiota and back pain is a novel area with ongoing research. A randomized trial study[11] showed that probiotics can reduce the proportions of *Peptostreptococcaceae*. Under normal circumstances, the gut microbiota, the host and the external environment establish a dynamic equilibrium, while the composition of gut microbiota are relatively stable. It is not easy to specifically target the abundance of a certain bacteria taxa, but it is more feasible to alleviate back pain by keeping a dynamic equilibrium of gut microbiota, with a relative decrease of *Peptostreptococcaceae* at the same time.

Thanks again!

5. Comments: Why did the authors perform a reverse MR analysis? What's the clinical motivation for identifying causal effects of back pain on microbiota?

Response: Thanks for your comments.

We conducted reverse MR analysis as a supplement to the primary results. We assumed that if there is a real causal relationship of microbiota to back pain, it is less probable to obtain the opposite causal relationship. It is a widely accepted strategy for MR analysis. Some other good Mendelian randomization articles also performed bi-directional analysis. For example, Liu et al performed bidirectional Mendelian randomization analyses on 3,432 Chinese individuals with whole-genome, whole-metagenome, anthropometric and blood metabolic trait data[12].

In addition, altered gut microbiota composition has been reported in people with back pain[5, 13]. From this observational evidence, we can not clearly identify the causal direction and totally exclude the possibility that back pain lead to microbiota changes. So we think it is necessary to apply a bi-directional MR approach. By comparing the results from bi-directional analyses, we can better elucidate the relationship of the two.

Thanks again!

Minor

1. Comments: There are inconsistent uses of "2-sample" and "two-sample" in the manuscript.

Response: Thank you for your helpful comments.

We are sorry for the confused places. Per your guidance, we corrected "2-sample" to "two-sample" in the abstract and discussion section.

Thanks!

2. Comments: The references for the GWAS summary data for gut microbiota are missing (Page 8, line 2).

Response: Thank you for your helpful comments.

We are so sorry for missing this reference. Per your guidance, we added the reference for the GWAS summary data for gut microbiota in the methods section. Thanks!

3. Comments: I'm assuming this is a typo, but on Page 8 line 5, the instrumental variables should be the genetic variants associated with gut microbiota.

Response: Thank you for your helpful comments.

We are so sorry for the confused description in the original manuscript. Per your guidance, we modified the improper description in the methods section. The gut microbiota was selected as instrumental variables (IVs) with a genome-wide threshold of 1×10^{-5} .

Thanks!

2 nd Peer Review	14-Sep-2023 to 16-Sep-2023
----------------------------

Reviewer's Responses to Questions

Please rate the importance compared to published work in this subject area.

Reviewer #2: Considerable - Top 30% in the subject area

Please rate the novelty compared to published work in this subject area.

Reviewer #2: Considerable - Top 30% in the subject area

Which aspects of scholarly presentation require improvement (if any)?

Reviewer #2: (No Response)

Do the methods, data and analysis (including statistical analysis where applicable) adequately test the hypothesis and support the conclusions?

Reviewer #2: Yes

Are the methods, data and analysis described in sufficient detail to be reproduced?

Reviewer #2: Yes

Where applicable, have the requested revisions been adequately addressed?

Reviewer #2: Yes

Second Editorial Decision 16-Sep-2023

Editorial Decision: Please make the necessary revisions by including the figures, figure legends, supplementary figures, and tables directly within the main text and supporting information. Ensure that all these elements are clearly presented and properly uploaded into the system.

Recommendation of the reviewers

Reviewer #2 Recommends Acceptance

Authors' response to 2nd Review
2023

19-Sep-

Update all the Figures and Tables properly.

Final Decision

20-Sep-2023

Accept the revised version for publication as the authors satisfactorily addressed the final comments of the editor.